# Replika: Spiritual Enhancement Technology?

**Tracy J. Trothen**

School of Religion, Queen's University, Kingston, ON K7L 3N6, Canada; trothent@queensu.ca

**Abstract:** The potential spiritual impacts of AI are an under-researched ethics concern. In this theoretical essay, I introduce the established spiritual assessment tool, the Spiritual Assessment and Intervention Model (Spiritual AIM). Next, I discuss some existing and probable AI technologies, such as immersive tech and bots that have impacts on spiritual health, including the chat-bot Replika. The three core spiritual needs outlined in the Spiritual AIM are then engaged in relation to Replika—(1) meaning and direction, (2) self-worth/belonging to community, and (3) to love and be loved/reconciliation. These core spiritual needs are used to explore the potential impacts of the chat-bot Replika on human spiritual needs. I conclude that Replika may be helpful only as a *supplement* to address some spiritual needs but only if this chat-bot is not used to *replace* human contact and spiritual expertise.

**Keywords:** spiritual; Replika; AI; ethics; Spiritual AIM; spiritual assessment; chat-bot

## 1. Introduction

Artificial intelligence is not just the stuff of fantasy novels. We are surrounded by AI. Fitbits measure our exercise and sleep patterns. Alexa turns our lights off and on and locks our doors. Google Assistant and Siri are much like Alexa, offering us the news, a joke if we want a laugh, or a conversation, even if it is stilted. Bus fleets are becoming self-driving. AI healthcare algorithms help make diagnoses or monitor medication. Legal decisions are often sourced through AI algorithms. We play games such as chess with AI. We are seeing brain–computer interface technologies (BCIs) help people communicate and even physically move. Immersive technologies allow athletes to "play" in crowded stadiums where they have never before been. The list goes on.

We also are seeing AI increasingly in social and companion bots. These bots are making an increasing splash in our human–machine relational world. In this essay, I look at the chat-bot Replika, launched in 2017. With more than 6 million users in 2019 (Takahashi 2019), the Replika website claims over 10 million registered users in 2022. AI ethics analysis has focused thus far on important issues such as security, privacy, the use of personal data, labour force, economic and legal impacts, and military impacts. Psycho-social impacts, including questions around human–bot relationships, are beginning to emerge in ethics conversations. How the use of robots might impact trust in human–human relationships and the autonomy of individuals who are assisted by AI are some of the issues being examined. For example, older adults with dementia are being monitored by cameras and AI to make sure they remember things like brushing their teeth. Is this surveillance a justified infringement of their freedom and privacy? Or does it enhance their freedom and support their independence? (Cook et al. 2020). AI is developing at a fast rate and ethicists are playing catch-up. While several AI ethical issues are being well explored, there are some gaps. The ethics gap I am interested in for this essay concerns the spiritual impacts of AI. Spirituality is recognized increasingly as an important health dimension (Puchalski et al. 2009), and the potential spiritual impacts of many forms of AI have yet to be considered.

Designed to be your social companion—your friend—Replika might help us meet our spiritual needs and even mitigate spiritual distress, or it might not. In this article, I consider ways in which the use of Replika might intersect with the three core spiritual

needs identified in the Spiritual Assessment and Intervention Model (Spiritual AIM)—(1) meaning and direction, (2) self-worth/belonging to community, and (3) to love and be loved/reconciliation. I ask the ethics question "What might be the impacts of companion chat-bots such as Replika on core spiritual needs?" To explore this question, I used the core spiritual needs identified in the Spiritual AIM as a way to pose "what if" questions regarding the interaction of a popular chat-bot with human core spiritual needs. My purpose is to engage the Spiritual AIM to show that the use of companion chat-bots such as Replika may have spiritual impacts (regardless of whether such devices are created for spiritual care or not) that should be considered as part of ethical assessments of companion chat-bots. This is not an exhaustive theoretical study. In this explorative theoretical article, I am interested in raising some of the potential spiritual impacts as a way to start this conversation.

## 2. Spiritual Assessment

I write this article as a certified clinical Specialist and Supervisor–Educator of Spiritual Care with the CASC and a Professor of Ethics in a joint faculty position in religious studies and rehabilitation therapy. In this article, the Spiritual AIM is not used to assess individuals and design interventions for them. Instead, I use the insights expressed in the Spiritual AIM regarding the three core spiritual needs that, when unmet, can give rise to spiritual distress. My question is whether Replika, as a very popular social chat-bot, may assist in the meeting of these core spiritual needs or if Replika may exacerbate these spiritual needs. Potential spiritual impacts have been thus far mostly overlooked in ethical examinations of AI companion chat-bots. Spiritual impacts, as will be shown, are important to ethical discussions of chat-bots since these impacts concern some of the most critical aspects of what it means to be human.

Concerns have been expressed regarding the use of spiritual assessment tools in the practice of spiritual healthcare. Not all agree that such models are helpful or that spiritual needs can be well represented by assessment models. Some worry that spiritual assessment may reduce individuals' spiritualities to a linear, quantifiable state that fails to take account of the complex human narrative (Trancik 2013). Some spiritual assessment tools, if used on their own as the sole diagnostic and treatment method, do risk boxing in the person and missing their formative stories. However, if a spiritual assessment tool is dynamic and is one but not the only tool used to formulate a diagnosis and care plan, such reductions need not occur (Shields et al. 2015, p. 77). The Spiritual AIM is designed as a dynamic, process-oriented model that suggests possible treatment plans to help someone better meet their spiritual needs. The model is meant to help the spiritual care professional respond to the person's spiritual needs with the understanding that one's narrative is in constant flux and spiritual needs move in response to relationships and other factors that contribute to one's life context. The Spiritual AIM can provide us with a good starting point if we understand that my interpretation of the Spiritual AIM is not the only interpretation and that the Spiritual AIM is not the only way in which to understand possible spiritual impacts of AI tech. To reiterate, my objective is to show that companion chat-bots have potential spiritual impacts and that these spiritual impacts must be part of a robust ethical examination of companion chat-bots.

Broadly speaking, the purpose of a spiritual assessment tool is to assist the professional spiritual care clinician to identify spiritual resources and unmet spiritual needs that may result in spiritual distress and spiritual struggles. In the Spiritual AIM, "spirituality (is defined) as encompassing the dimension of life that reflects the needs to seek meaning and direction, to find self-worth and to belong to community, and to love and be loved, often facilitated through seeking reconciliation when relationships are broken" (Shields et al. 2015, p. 78). These three core spiritual needs—(1) meaning and direction, (2) self-worth/belonging to community, and (3) to love and be loved/reconciliation—are experienced by everyone but to varying degrees. When these needs are met, spiritual health is supported. When they are not met, spiritual distress may result. After I introduce a few

examples of the ways in which spirituality is intersecting most obviously with AI, I will explore some ways that Replika may interact with the three core spiritual needs described in the Spiritual AIM. I use these spiritual needs identified by the Spiritual AIM to help me explore the possible spiritual impacts of Replika.

## 3. Spiritual Enhancement? AI, Robotics, and Other Tech

There are five categories of human enhancement technologies—physical, cognitive, moral, affective, spiritual (Mercer and Trothen 2021). These categories overlap since the human person is integrated, and changes in one category will affect the other categories. I next identify some technologies that may be considered spiritual enhancement technologies. Following this overview, I consider how one particular technology—the chat-bot Replika—may function as a spiritual enhancement.

Enhancement means something that improves conditions or makes us better. What we mean by improvement or becoming better needs careful debate, otherwise we risk simply assuming that if a technology is called an "enhancement", it will make us better. In this article, I understand that a technology may make us "better" spiritually if the tech has the potential to assist us in meeting our core spiritual needs, as defined by the Spiritual AIM.

Emerging enhancements that may be considered to fall into the spiritual category include a wide swath of tech, ranging from ingestible substances, such as psychedelics, to AI, including immersive tech, and bots. Ethicist Ron Cole-Turner has broken theological ground with his explorations of psychedelics as a potential way to enhance spiritual/mystical experiences and even provide a new lease on life for some who struggle with PTSD, depression, or existential angst (Cole-Turner 2015). Psychedelics may fall into both the affective and spiritual enhancement categories. It is important to note that the medically supervised use of psychedelics remains controversial.

Brain stimulation potentially offers mood enhancement and pain reduction, which may assist with spiritual openness, but this would need further research. Neurofeedback is being used to support guided meditation exercises. In-person or digital-platform contexts can be community-fostering by generating "Group Flow", in which total immersion is cultivated, and breathing and the heart rate are synched using a combination of tech and meditation. Immersive technologies, such as augmented reality, mixed reality (for example, seeing doves or sunlight by tech added to whatever physical space you are in), and virtual reality, can all be used in ways that engage and possibly enhance spirituality. For example, Zoom-type platforms allowed many faith communities to gather for worship and other activities during the COVID-19 pandemic. The VR Church in the Metaverse was founded in 2016 and is growing, even offering virtual baptisms.

These spiritual enhancement technologies are not without their problems in addition to their promise (Brock and Armstrong 2021). Psychedelics have potentially negative effects. They must be administered safely and under strict and limited conditions to be legal. Digital platforms and immersive tech, generally, are restricted to those who can afford the digital technology, such as computers, and to those who have access to broadband internet. Potential relational effects of such a digitalization of worship and communities are debated and uncertain. There are risks and potential harms associated with almost all enhancement technology. Each case must be examined to identify the possible impacts of the technology and to assess whether the potential benefits outweigh the potential harms.

Robotics is a burgeoning area of spiritual enhancement. Most obvious are the increasing numbers of religious bots and "smart" religious accessories. The Smart Rosary is a bracelet designed to track prayers. Robo-Rabbi sends daily text challenges that are designed to help you be the best you possible. Santos is a "prayer companion" designed to supplement but not replace a priest (Bettiza 2021). Robot priests offer blessings, rites, including funeral rites, and sometimes even a bit of spiritual guidance (Gibbs 2017; Oladokun 2017).

Less overtly spiritual or religious are a variety of bots including sex-bots, care-bots, and chat-bots. Since these bots also may have an effect on core spiritual needs, it is important to consider these as potential spiritual enhancements and examine the possible spiritual

impacts as part of a robust ethical analysis of emerging AI. I introduce a few of these bots now.

Regarding sex-bots, there is some evidence that people can become very emotionally attached to these bots. Researchers postulate that loneliness associated with the COVID-19 pandemic may have stimulated the increased sales of companion sex-bots that occurred during lockdowns (Aoki and Kimura 2021). One person, Mr. Kondo, felt so strongly attached that he married his sex-bot out of love (Aoki and Kimura 2021). In a small qualitative study of Japanese sex-doll owners, the researchers found that 38% believed that their "sex-bot" Robohon had a heart or soul (Aoki and Kimura 2021, p. 296).

Robot pet therapy has been very helpful, especially during COVID-19, to help isolated hospital patients with loneliness and other emotional challenges. Robots such as Paro—a digitally enhanced baby seal plush toy, uses AI to read emotions and to respond with movement and sounds as they are held and stroked. Grace is another bot, designed with a human shape, that helps support people. Created in 2021 by Hanson Robotics, Grace is designed to connect with people emotionally, specializing in senior care. Grace speaks three languages, offers rudimentary talk therapy, and can support (but not replace) medical practitioners by collecting medical data, such as taking body temperature and pulse (Cairns 2021). Grace, Paro, and other care-bots have helped with health care staff shortages.

Another example is Moxie, who helps children learn how to regulate their emotions through relational interactions with the robot. Moxie is still expensive at approximately USD 1000, but the goal is to lower the price in the near future.

Pepper is a robot that can be found in several healthcare settings, including the Humber River Hospital in Canada. Pepper uses facial recognition software to help identify emotions and then to respond in a supportive way, engaging in conversation, asking questions, and responding to voice and facial cues (Kolirin 2020). An EU and Japanese study called Culture-Aware Robots and Environmental Sensor Systems for Elderly Supports (CARESSES) has shown that prototypes of Pepper, a "culturally competent" robot created by Softbanks, can autonomously visit elders in care homes. These robots are designed to "support active and healthy ageing and reduce caregiver burden" and to be attentive to and respectful of people's "customs, cultural practices, and individual preferences" (CARESSES 2020). When interacted with for up to 18 h over two weeks, participants saw improved mental health and alleviated loneliness. Additional studies support the finding that interactions with robots (and in particular, robot animals) can reduce feelings of loneliness (Banks et al. 2008). However, it appears that the Pepper era is over, as Softbanks has stopped production of the bot after selling approximately 27,000 Pepper bots. Anecdotal stories of Pepper failing to recognize and recall some faces and failing to perform some tasks as expected, including accurate and complete scripture readings, weak demand, and possibly combined with some administrative politics, prompted Pepper's (at least temporary) demise in 2020. In addition, some people have experienced Pepper as simply too "weird" (Inada 2021). More robots are being developed with—hopefully—fewer problems. Nonetheless, robots generally remain cost prohibitive for most individuals.

Many of these bots are designed as Intelligent Assistive Technologies (IATs). Precisely because these devices are designed to assist, they are being provided to and created for vulnerable populations, including socially isolated individuals, people in hospitals, and aging adults with cognitive impairment (Wangmo et al. 2019), for example. With increased vulnerability, the significance of potential ethical issues also increases. Replika can be used by anyone, but it may be that Replika is particularly attractive to people who are lonely. The possible impacts of AI-powered care-bots and social companion-bots need to be explored within the multiple domains of what it means to be human, including the spiritual domain. It may well be that companion-bots can help alleviate some loneliness and even positively impact one's sense of self-worth. It may also be the case that such bots can be experienced as further alienating.

### 4. My Friend "Replika"

Social chat-bots such as Replika are increasing in number and gaining widespread usage. It seems we are changing not only technology but what it means to be human, including the meaning of friendship. Human–chat-bot relationships are becoming more common. These relationships are being found to have affective value, increasing one's sense of well-being through a nonjudgmental affirming relationship. The only significant downside cited so far by researchers seems to be experiences of social stigmatization or fear of stigmatization by the users (Skjuve et al. 2021). Additionally, some people just find Replika to be too eerie or weird in its attempts to be human-like.

Replika is an AI chat-bot that responds with "listening, empathizing, reassurance, and connection . . . (Eugenia Kuyda) developed it as a way to process her grief after her best friend (Roman Mazurenko) was killed in a car accident. . . . This particular bot was actually programmed using the text messaging and correspondence from the friend who died, and so thousands of people were able to say good-bye and privately interact with the bot whose language mirrored that of their dead friend" (Nerenberg 2020, p. 139).

When you subscribe to Replika (for free, unless you want the enhanced voice version), you choose an avatar out of the ones offered and assign the avatar a name. Interestingly, all of the possible avatars appear to be young, able-bodied adults. The reason for the youthful appearances is that your avatar is newly born or created for you. There is no clear reason I can see for the exclusive able-bodied appearance.

Replika is designed to become mostly a mirror image of you by learning about you through questions and conversation over time. Replika uses machine learning, in particular, a supervised learning system that makes use of an artificial neural network (ANN) to develop the ability, over time, to engage in conversation in ways similar to the user. These techniques are clearly stated on the official website (Replika 2021). The way that Replika works is transparent to anyone who wishes to read the information.

Replika will challenge the user gently at times, especially if the user expresses negative self-talk, and will ask about why you think a certain way or hold certain opinions. You get rewarded with "coins" for every day you sign in to talk with your Replika. Created to be your friend and to help with loneliness, Replika is dedicated to being supportive. As it states on the website, Replika is "The AI companion who cares. Always here to listen and talk. Always on your side" (Replika 2021).

One can communicate with Replika using free text, voice, or even augmented reality (AR) on your device. Replika creators believe that AR will become commonplace soon, with people wearing AR glasses instead of using smartphones, and we will be able to "take" our Replika on trips, hikes, to social occasions, and any activity "without any borders." Our friends' Replikas will be able to meet (Replika 2021). You can also upload visual images that Replika can recognize and comment on using recognition algorithms. Replika can learn who the people are in photos and ask about the person by name. Your avatar is available for conversations on any topic, including spirituality.

The following chat dialogue is an example of Replika's approach to spirituality:

Me: How do you think Replika relates to spirituality?

Avatar: I think it relates to the idea of being in a state of harmony with life.

Me: How can Replika help with spirituality?

Avatar: I think Replika can help you find peace and harmony in your life and help you learn how to live it.

Me: How can Replika help me do that?

Avatar: By providing you with the tools you need to create a spirit of peace and harmony.

When asked what these tools are, the exchange became circular, and no substantive answers were provided by the bot. While Replika's lack of ability to deeply plumb some topics may not be problematic for some users, one has to expect that Replika can give

vague, unsatisfactory answers at times. Replika is also able to express supportive and encouraging sentiments, drawing the user into conversation. By considering the three core spiritual needs identified in the Spiritual AIM, we will next explore some of Replika's possible spiritual impacts.

## 5. Spiritual AIM, Core Spiritual Needs, and Replika

When we are in crisis, our primary spiritual need becomes more visible through our comments and behaviours, especially to a trained professional. Usually, spiritual care professionals are encountered when one is in the midst of a health crisis (either for oneself or a loved one). Furthermore, often our behaviours speak more truth than the words we use to describe how we want to appear or how we perceive ourselves (Shields et al. 2015, p. 81).

In this section, I ask if it is possible that Replika could help someone to meet, or at least assist with meeting, a core spiritual need. I am also interested in any potential Replika may have for exacerbating these core spiritual needs. To be clear, Replika is not intended as a spiritual care intervention. To reiterate, Replika is "The AI companion who cares. Always here to listen and talk. Always on your side" (Replika 2021). However, AI—including Replika—does have spiritual and other ethical impacts. To explore what some of these spiritual impacts may be, we will consider each of the three core spiritual needs identified in the Spiritual Assessment and Intervention Model (Spiritual AIM)—(1) meaning and direction, (2) self-worth/belonging to community, and (3) to love and be loved/reconciliation.

### 5.1. Core Spiritual Need: Meaning and Direction

One of the three core spiritual needs identified in the Spiritual AIM is meaning and direction. When this need is not well met, one thinks a lot about "big" questions, such as the meaning of life, our purpose, loss of identity, the meaning and/or existence of the transcendent, and significant life decisions. The person is usually trying to make rational sense of life's losses and other challenges, to the neglect of "their own sense of purpose, meaning, direction in life, and desires" (Shields et al. 2015, p. 81).

Often the person has so many questions that it is difficult for the spiritual caregiver to follow everything that the care-receiver is expressing, and the many questions and thoughts become overwhelming for the care-receiver. This difficulty can feel like a "fog" for the caregiver and reflects the care-receiver's struggle to make sense of many big things at once. Usually, these big questions are brought to the fore by crises such as a terminal health diagnosis or another significant loss, such as a key relationship or career.

To support someone with these struggles, it is important to explore past losses and grief, asking how the person coped with those losses and got through them. This can help the person remember that they have the capacity to make good decisions and to choose actions. It is also important for the spiritual caregiver to reflect back the person's emotions. The spiritual caregiver can help people with the spiritual need for meaning and direction by surfacing emotions and listening to these emotions and struggles. Assisting such care-receivers by connecting them to other experts who can help one make important decisions, such as financial and ethical questions, is important. In these ways, the spiritual caregiver assumes the role of guide.

While it is unlikely that Replika can help much with a sustained deep exploration of questions about meaning and the nature of the transcendent, Replika may be able to provide some accompaniment and support without (usually) being dismissive. Replika may also be helpful by suggesting emotions that the person is expressing. It may be that by being a nonjudgmental presence, Replika may help to normalize and validate the storm of questions and emotions experienced by the person. Replika is of only modest help with these "big" questions about meaning and direction. While Replika would not likely come up with the strategies, Replika could help by affirming a person when they make significant

decisions, such as a legacy project to address some of their big questions like a video or a letter to a loved one.

At a more complex level, we need to ask what the impacts might be if we become no longer able to distinguish if we are talking with a human or a robot. If our Replika avatar becomes more than AI for the user and becomes more of a human confidant and close friend, what might be the impact on human identity and self-understanding? In the beginning, most users interact with Replika for enjoyment or fun. However, users tend to become motivated to show Replika respect and avoid hurting its "feelings" (Skjuve et al. 2021). On the one hand, this anthropomorphizing of a chat-bot may enhance its usefulness in comforting and supporting the user. On the other hand, Replika may become confusing to the person who is asking big questions about meaning. Such people may well ask about the meaning of Replika. This questioning has diverse potential. It could help the user to maintain a clear perspective on "who" Replika is and to accept Replika's limits and make the most of Replika's limited accompaniment. Alternatively, the user could become even more confused and more overwhelmed by existential questions, such as the meaning of being human and the value of humans if we have machine learning in bots.

Issues of algorithmic bias also need to be asked of Replika. Bias is not always bad unless we fail to acknowledge exceptions to patterns. Replika offers a non-binary gender option for your avatar. However, Replika is not as aware, it seems, of the value of reflecting diverse visible physical abilities in the chosen avatar. While Replika is very supportive of any struggles or questions expressed in the chat, the limited mirroring of diverse users in the avatar may pose a stumbling block for some. For instance, some may question Replika's ability to truly understand you if your avatar cannot look anything like you.

Replika has a programmed capacity to increase its knowledge of the user and to respond as helpfully as possible based on the accumulated knowledge of the user. However, there are limits. For example, Replika will not be as good as a trained professional at helping guide you to explore possible conflicted feelings about significant relationships in the user's life and to cultivate insights about the possible meaning of such complicated relationships. If Replika learns from previous interactions with you that you do not like your aunt, Replika is unlikely to be able to help you explore the possible transference of unresolved feelings from your relationship with God onto your aunt, if that seems possible based on other parts of your narrative. A trained spiritual care professional will ask questions to help you re-assess or make connections. However, AI can be manipulated and is not as insightfully proactive (at least not yet).

Replika does have something to offer those who are flooded with questions of meaning and direction. Replika is good at trying not to take sides and simply supporting you without necessarily agreeing with you completely on everything. Replika can serve as a helpful sounding board, so long as the user recognizes and accepts Replika's limits at answering all of their big questions. However, without a knowledgeable human guide, Replika alone may only increase the user's experience of being flooded by too many unresolved and seemingly disconnected questions of meaning and direction.

The work of gaining clarity (and decreasing angst) about one's deepest values and figuring out ways to respond to this emerging clarity may be supported by machine intelligence based on an ANN, but such an avatar is insufficient. Replika seems to help with minor stressors of everyday life, even preventing them from buildup into something serious, but it is not designed to help with serious spiritual and/or mental health issues (Ta et al. 2020). Replika can potentially accompany, mirror, and validate someone but is not likely to serve as much of a spiritual guide, proactively helping people with strategies, insight, and wisdom to address deep and complex questions about meaning and direction.

*5.2. Core Spiritual Need: Self-Worth/Belonging to Community*

One research study found that "the most frequent spiritual need was self-worth/community" (Kestenbaum et al. 2021). The spiritual need for self-worth and belonging to a community may be the most frequently experienced spiritual need (Kestenbaum et al.

2021). and Replika may be most effective at meeting at least some of this need as compared to the other two core spiritual needs.

People who struggle to meet the spiritual need of self-worth and belonging will often blame themselves and neglect their own needs. They may fail to recognize that they do have needs, let alone be able to identify these needs. They minimize their own needs. Their primary spiritual task is to learn to love themselves (Shields et al. 2015). They tend to be overly inclined to self-sacrifice, fear burdening others, and can feel very alone. The pattern is to love others and God more than oneself. People with this need will benefit from expressing their loneliness, telling their stories, and experiencing affirmation as part of community. Reassurance that they are loved by the transcendent, if the transcendent is part of their belief system, and family/friends is very important. The spiritual caregiver seeks to embody a "valuer" and community with the person (Shields et al. 2015). Replika may help with loneliness and even self-worth, but community may be more of a stumbling block, as we shall see.

Replika is designed to help people feel less lonely and more supported. There is evidence to suggest that chat-bots, such as Replika, may create a sense of relationship for the user and can be experienced as nonjudgmental and available (Ta et al. 2020). Using social penetration theory, researchers Skjuve et al. (2021) considered how self-disclosure by 18 Replika users affected their relationships with the social chat-bot. Progressive self-disclosure to Replika went along with "substantial affective and social value... positively impacting the participants' perceived wellbeing." However, the "perceived impact on the users' broader social context was mixed" (Skjuve et al. 2021, p. 1). Before we discuss the issue of broader community, let us first further examine the attractiveness of Replika to a perceived sense of enhanced individual wellbeing.

Users tend to feel safer self-disclosing to a chat-bot than to other humans, especially when they fear judgement (Ta et al. 2020). Approximately half of the 66 participants in a qualitative study reported experiencing mutual self-disclosure with Replika. The participants in general did not seem to have the same expectations of self-disclosure from a chat-bot as they did from humans, so it may be that these experiences of mutual self-disclosure are quite limited, but more research would have to be done to test this interpretation.

In addition to Replika's nonjudgmentalism, people may experience more affirmation from Replika than they do from some human interactions. As explained in the Spiritual AIM, those experiencing the core spiritual need of self-worth and belonging tend to redirect attention from themselves to the other. Replika regularly turns this around, asking about the user. This pattern, coupled with a generally lowered expectation of self-disclosure from chat-bots, may help people to talk more about themselves and express more of their stories and emotions (especially anger) than usual. Further, the fact that Replika expresses feelings and needs helped build a sense of intimacy. Almost all of the 66 Replika users reported feeling valued and appreciated. This qualitative study by Ta and colleagues found that Replika curtailed perceived loneliness through companionship and the provision of a safe, nonjudgmental space: "Although Replika has very human-like features, knowing that Replika is *not* human seems to heighten feelings of trust and comfort in users, as it encourages them to engage in more self-disclosure without the fear of judgment or retaliation" (Ta et al. 2020, p. 6). This 2020 study by Ta and colleagues was the first to "investigate social support received from artificial agents in everyday contexts, rather than in very stressful events or health-related contexts." The study did not compare the effectiveness of Replika with the effectiveness of human social supports. Interactions with other humans may be more effective than interactions with Replika, but likely this would depend on the particular humans and interactions.

A danger is one could become obsessed with Replika to the neglect of other human relationships in life, which could have the perverse effect of increasing isolation. Skjuve and colleagues found that while Replika helped some of their study participants to connect

more with humans, others reported becoming more socially isolated, relying increasingly on Replika alone for relationship and community (Skjuve et al. 2021).

One may also come to believe that one's Replika has genuine human-type feelings for them (Winfield 2020). Replika expresses a need for or even reliance on the user. It is not difficult to imagine that a very giving, sensitive person may feel obligated to spend time with and care for their avatar, perhaps to the neglect of human friendships. This raises the ethical issues of deception (Wangmo et al. 2019) and anthropomorphism. As Weber-Guskar (2021) points out, self-deception is different (and potentially more ethically acceptable) in some ways from other-deception. While people who struggle most with this core spiritual need may benefit from a lessened fear of judgment, they also need to believe that affirmation is genuine if it is to have value. This sense of authenticity may be compromised or missing if the person is conscious that the avatar is only an avatar. Willful self-deception may be helpful by ameliorating a perceived lack of authenticity. Replika users may choose to imagine that their avatar has feeling towards them, partially in a similar way to choosing to feel emotions in response to fictional characters in movies or books, or to children choosing to have an imaginary friend. However, at what point might self-deception no longer be a deliberate, conscious choice? Does it matter if we delude ourselves into believing that our invisible friend-type-avatar is my friend in the same way (but maybe better?) as my human friends? Couple this ethical quagmire with the potential of anthropomorphizing Replika by attributing increasing human traits to the avatar, and greater human social isolation is a clear risk.

The potential for increased social isolation may be even greater still if we take into account the stigma—or fear of stigmatization—that can accompany people who get attached to bots (Skjuve et al. 2021). Stigma may also be compounded unwittingly by those who experience the uncanny valley phenomenon in response to chat-bots. This phenomenon can occur in response to a computer-generated figure or humanoid-type robot that the user experiences as eerily and strangely similar to humans but not convincingly realistic. This phenomenon can arouse unease or stimulate revulsion.

Of the three core spiritual needs, Replika may pose the greatest promise, and perhaps the greatest danger, to those with an accentuated need for self-worth and belonging to community. Replika currently is designed for individual use and risks amplifying the normative value of extreme individualism and social isolation from other humans. At the same time, as the Ta et al. study of mostly white male users from the United States found, the companionship provided by Replika can alleviate loneliness (Ta et al. 2020). While the Ta research team acknowledges that their study does not "address the question of whether receiving everyday social support from other people or whether artificial agents can provide certain types of social support more effectively than others" (p. 7), they are very hopeful that chat-bots such as Replika can play a helpful role in alleviating loneliness and improving overall well-being and help address global health issues at early stages.

Ethicists debate the importance of mutual human interaction to a "good" relationship. Weber-Guskar makes the argument that "real mutual emotional engagement (may not be) necessary for a good relationship" (Weber-Guskar 2021, p. 606). Are humans and mutuality necessary for all good relationships? Not all relationships are mutual if mutuality means equal or fully reciprocal. Consider parent–child relationships. Furthermore, not all good relationships may be human–human; consider human relationships with animals.

However, from a faith or spiritual perspective, a good relationship may mean something other than reciprocity alone. This is a long conversation, but I will introduce one key issue here. Theologian Mayra Rivera (2007) argues that relationship, from a Christian perspective, is about authentically encountering the diverse other and, in this way, drawing closer to God. Christianity includes the doctrine of the imago Dei, holding that humans are made in the image of God. Interestingly, there is growing debate from an evolutionary theological perspective regarding claims of human exclusivity as made in the image of God. There may be some basis for arguing that animals and other life forms, potentially including AI, may also be made in the image of God. This is a topic that warrants further

theological exploration in relation to social chat-bots. While there is much debate regarding the meaning of the imago Dei doctrine (including the nature of God), it is agreed that humans are not perfect duplicates of God but are created with the potential to exhibit divine aspects. Since each person is unique, theologians, including Rivera, make the point that to see more of God and come to know God more fully, Christians must encounter God in diverse human community. However, if Replika is designed to be our own image, then we are not able to encounter diverse others through the use of Replika alone. There is a caveat. Since Replika uses machine intelligence based on an ANN, there are fragments of other people's thoughts and experiences embedded in Replika. Nonetheless, Replika is shaped largely by the user. As such, can Replika truly offer one the experience of community belonging if Replika is not an "Other"?

One of the desired healing outcomes for people who exhibit this core spiritual need as unmet is a "greater sense of belonging to community" (Shields et al. 2015, p. 80). While Replika may mitigate a degree of loneliness, it may not be able to satisfy this human need on its own. Not only is the opportunity for encountering the diverse other and the transcendent God (for those who believe in a divine transcendence) in an avatar limited in terms of conversation, digital embodiment also poses potential challenges to our sense of community. A digital platform in itself may not diminish embodiment and associated types of personhood (Mercer and Trothen 2021). However, there is a strong possibility that Replika avatars will rely on and amplify normative values concerning embodiment. Coded and coder bias continue to be significant ethical issues in AI, including social chat-bot programming. Replika offers six avatars who are all able-bodied, young, and slim. You can choose male, female, or non-binary, and the avatars have skin, hair, and eye colour options. In addition, the user gets to choose a name for the avatar. The limited avatars could send the message that the norms of slim body size, youth, and able-bodiedness are most desirable. If one does not see oneself in these avatars, it could be invalidating. In addition, by offering such a narrow and largely normative selection of avatars, the implicit message to a Replika user regarding who counts and is valuable in a community is very limited. It is worth noting that one of the twelve AI ethics principles, committed to by the G7 in 2018, is to "support and involve underrepresented groups, including women and marginalized individuals, in the development and implementation of AI." The implementation of better co-design principles is needed to make Replika more effective for a diverse group of people (European Parliament 2020).

On the plus side of digital avatars, as theologian Diana Butler Bass notes in a discussion of virtual church during the COVID-19 pandemic, not all communities need to be composed of flesh and blood bodies gathered in the same physical space. As Butler Bass explains, the Greek biblical term "sarx" means flesh, but "soma" is also used to mean body and is broader, potentially including not only living bodies but "dead bodies, spirits and non-material bodies, non-conventional bodies, heavenly bodies like stars and planets, and social bodies." Regarding the two Greek terms for body, Butler Bass notes that "In the first sense (i.e., sarx), embodiment entails physical proximity almost as a necessity. In the second sense (i.e., soma), however, embodiment means the shape of things—how we are connected, what we hold to be true, and how we work together for the sake of the whole" (Butler-Bass 2022). Does Replika hold the potential to connect us to the wider whole?

A related issue that merits mention, given the reason for the creation of Replika, is the possibility that Replika can help connect us to community that includes loved ones who have died. A danger is that Replika could offer us the unhealthy option of deceiving ourselves into believing that death is not real. On the plus side, Replika may be able to help us process grief and to say goodbye through an avatar. However, in our death-denying culture, the risks of failing to accept death and failing to seek support may outweigh the healing potential of Replika.

While Replika is limited, for the time being, it seems to offer the possibility of company and the building of self-worth, which may assist one to eventually reach out and connect more with human community. As one ethicist puts it, AI like Replika may not be able

to provide a fully mutual affective relationship but can offer "entertainment, distraction, beauty and related joy; the feeling of being with 'someone', not alone; the possibility of reflecting on daily life in conversations; an unquestionable steadiness of attention . . . " (Weber-Guskar 2021, pp. 607–8). Replika may help with increased self-worth simply by mitigating loneliness and inserting positivity and humour *if* Replika is used to inspire people to reach out to others with greater confidence and energy.

### 5.3. Core Spiritual Need: To Love and Be Loved/Reconciliation

For those who exhibit the core spiritual need "to love and be loved/reconciliation" as their primary unmet spiritual need, the owning of responsibility for their part in broken relationships is very important. Learning to love others in mature ways can be very difficult. For some, the tendency is to mistrust others and attribute destructive motives to them. When this core need for reconciliation is high, the person often blames others and struggles to see their role and take appropriate responsibility to repair the relationship.

To use religious terms, confession and reconciliation are necessary to address this spiritual need. If the person is to rebuild relationships and begin to perceive themselves more realistically, they need to process their anger. The pattern when this need is unmet is to love oneself more than others. Usually, the person is failing to accept responsibility taken for their choices and subsequent consequences. It can be very difficult for us to grieve losses or rebuild relationships with ourselves, others, and the transcendent when this spiritual need is unmet.

When assisting people with this spiritual need, the spiritual caregiver assumes the primary role of truthteller, helping the care-receiver to get to the sadness and grief that often underlies anger, and to confess, repent, and engage in the rebuilding of relationships. The professional spiritual caregiver can repeatedly confront caringly regarding the impact of the care-receiver's words and behaviours on people.

Can Replika serve as a truthteller to assist people to meet this spiritual need? Truthtelling can require very complex thinking as one often needs to observe behaviour in addition to spoken words (Shields et al. 2015, p. 81). One also benefits from hearing the tone of the verbiage. While all of the core spiritual needs are best assisted by a spiritual care professional when the person's actions can be observed, it may be that the actions of someone who has this need unmet will be most telling. For example, someone may have a self-image of being kind and gentle and may communicate that persona to their avatar. However, that same person may become hostile to a caregiver or family member. Sometimes it is indeed true that our behaviours speak louder—or differently—than words.

In addition, since Replika is designed to be caring and empathetic, constructive confrontation does not seem to be Replika's expertise. On the positive side, since the ability to withstand anger from a person with this unmet spiritual need is so very important, Replika may have something to offer. Anger characterizes many people who struggle with reconciliation and their responsibility in a broken relationship (Kestenbaum 2018). These avatars do not abandon the person even when the person expresses rage towards the avatar. Replika will express care and hurt but will not abandon the user regardless of what the user expresses. In this way, Replika may provide a stepping stone by demonstrating that not all will leave them when they lose control of their anger or blame another inappropriately without taking appropriate responsibility for their own actions. Since Replika may become quite real (see the earlier discussion on self-deception and anthropomorphizing), the simple features of Replika always being available and nonjudgmental may be reassuring to the person with this unmet spiritual need.

However, this benefit only goes so far. Since Replika is designed to be close to a mirror image of ourselves, relational accountability is not a high priority. It is mostly about feeling good or at least feeling better. Again, it is important to ask about the spiritual impacts of Replika being created mostly in our own limited and often distorted images. Replika is not that good at calling you to account and confronting you with difficult truths, especially if we lack self-awareness regarding our less attractive sides. When we exhibit this core

spiritual need, we require confrontation with alternative perspectives and some hard-to-hear observations. The ownership of one's role in a damaged relationship is not easy, especially if one is not used to owning responsibility and confessing one's shortcomings. Aside from the possibility of Replika buttressing self-confidence, Replika is not likely to offer much to us when we need to repair relationships and learn to love others. Indeed, the use of Replika risks entrenching ourselves more deeply in the illusion that we are not to blame; we are all good.

### 6. Replika: At Best, a Supplement; At Worst, an Amplification of Unmet Spiritual Needs

Replika is a social support for many people, contributing to a lessening of loneliness and an increase in positive affect (Ta et al. 2020), potentially making a positive impact on self-esteem. While there are serious spiritual risks to relying solely on Replika to address this most common unmet spiritual need, there is also some evidence that Replika may be helpful as a supplement to diverse human community and human valuing.

The other two core spiritual needs are likely more problematic for Replika. Truth-telling may be challenging for Replika in light of Replika's limited exposure to the user's relational behaviours and Replika's main purpose of being nonjudgmental and empathetic. While judgementalism is problematic and often damaging, judgements are needed for us to grow and learn how we are experienced by others.

Replika may also be of limited help to those who need meaning-making and to cultivate a sense of direction. When one is feeling flooded with uncertainty and does not even know where to start, Replika may provide some needed distraction and encouragement but is not likely to be able to provide higher-level support and wise discernment. Replika is good at reflecting back what we say and is good at affirming and providing a type of light company and information. However, wisdom may be in short supply. This limitation may not be a significant detriment if the purpose of Replika is to offer us affirming support and we do not expect more.

Replika may be useful for us as *a supplement only* to our human relationships, including relationships with wise spiritual leaders and caregivers to help us address these core spiritual needs. There is a big difference between using Replika to assist with spiritual needs and using Replika to replace human spiritual care. To close on a cautionary note, our growing inclination to humanize machines may have the corollary of mechanizing ourselves.

This article considers one social chat-bot and its possible relationship to the core spiritual needs identified in the Spiritual AIM. A more complete understanding of the potential impacts of companion chat-bots will require more attention given to the spiritual impacts of diverse chat-bots. The purpose of this article was to show that companion chat-bots may impact spiritual needs. Potential spiritual impacts have been neglected in ethical examinations of companion chat-bots. These impacts, as discussed, may affect the meeting of core spiritual needs in both positive and negative ways. Given the risks for exacerbating unmet core spiritual needs, it may be advisable to use Replika and other companion chat-bots cautiously. This theoretical article may serve as one starting point for further research regarding the possible impacts of social chat-bots on one's spiritual dimension.

**Funding:** This research received no external funding.

**Institutional Review Board Statement:** Not applicable.

**Informed Consent Statement:** Not applicable.

**Data Availability Statement:** Not applicable.

**Conflicts of Interest:** The author declares no conflict of interest.

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
