# Peer review of "Replika: Spiritual Enhancement Technology?"

_religions, doi:10.3390/rel13040275_

Round 1

Reviewer 1 Report

This is an engaging essay exploring one piece of contemporary technology and its possible implications for spirituality and spiritual development that readers will find stimulating and informative. Only one typo -- line 64 "cn" should be "can"

Author Response

I am grateful to peer reviewer #1 for catching this typo and for the encouragement. I have corrected this typo. 

Reviewer 2 Report

Thank you for the privilege of reviewing this paper. I found it very interesting and relevant to emerging realities. I have only very minor comments for suggested improvement (most of them simply typos).

Line 40 – typo – spirituality

Line 64 – typo – can

Line 84 – Is there a word missing at the beginning of this sentence?

Line 159 – check wording

Line 170 – incomplete sentence

Lines 211-212 – inconsistent spelling of stigmatization

Line 341 – at least not yet

Line 391 – missing a space

Line 393 – spiritual

Line 422 – struggle

Lines 516-18 – not sure I understand this sentence. The following sentence is also not quite clear enough.

Line 534 – Do you mean to say “Learning to love others can be very difficult”? Not quite clear.

Line 616 – named

The list of references requires attention to consistent format.

Author Response

I am grateful to peer reviewer 2 for catching these minor mistakes. I have corrected all of those minor points.

Thank you for such a careful read of my essay!

Reviewer 3 Report

The article is clearly written and draws on established research in the areas of both AI and spiritual care. It also feels a bit like a setup. First, it evaluates Replika for a purpose that the bot is not intended to serve; the author does admit this but then proceeds since companion bots, AI in general, are becoming so engrained in the broader culture. Second, it assumes the validity and reliability of the assessment tool, Spiritual AIM, which when combined with the first point makes it seem as if the purpose of writing the piece was to promote Spiritual AIM. This sense is reinforced by the continued self-referencing of sources. 

The author, to his/her credit, offers a good many qualifications, and thus paves the way to offer tentative conclusions about a device that may or may not be used for the purposes of spiritual care. It makes a scholarly contribution within the narrow parameters established. And this is no doubt in keeping with the overall theme of the issue under development. The article adds to the conversation, and it does so in a professional manner. 

That said, one might consider ways in which someone deploying Spiritual AIM as a spiritual care tool could come across much like the very bot-like device that is Replika. Armed with assumed patterns of what constitutes spiritual health, the spiritual caregiver plugs and plays the "three core spiritual needs." This article itself reads like a routinized acting out of spiritual care...directed at a bot unable to defend itself. 

Author Response

The comments from Reviewer #3 helped me to re-examine my presentation of the Spiritual AIM Model. My purpose is to engage the Spiritual AIM Model to demonstrate that the use of companion chat-bots such as Replika have potential spiritual impacts (regardless of whether such devices are created for spiritual care or not) that should be considered as part of ethical assessments of companion chat-bots. I have inserted a few sentences and deleted a few others to try to make this purpose clearer, and to address the reviewer’s concern that I may be promoting this assessment tool too strenuously. I also included a stronger emphasis on the dynamic and process oriented purpose of Spiritual AIM as a model for assisting in spiritual care.

I am very grateful for the reviewer's comments. The essay is much clearer now.